# Predicting the Potential Geographical Distribution of *Rhodiola* L. in China under Climate Change Scenarios

**DOI:** 10.3390/plants12213735

**Published:** 2023-10-31

**Authors:** Meilin Yang, Lingxiao Sun, Yang Yu, Haiyan Zhang, Ireneusz Malik, Małgorzata Wistuba, Ruide Yu

**Affiliations:** 1State Key Laboratory of Desert and Oasis Ecology, Key Laboratory of Ecological Safety and Sustainable Development in Arid Lands, Xinjiang Institute of Ecology and Geography, Chinese Academy of Sciences, Urumqi 830011, China; yangmeilin@ms.xjb.ac.cn (M.Y.); sunlx@ms.xjb.ac.cn (L.S.); hyzhang@ms.xjb.ac.cn (H.Z.); irekgeo@wp.pl (I.M.); malgorzata.wistuba@us.edu.pl (M.W.); ruideyu@ms.xjb.ac.cn (R.Y.); 2Faculty of Earth Sciences, University of Silesia in Katowice, 41-200 Sosnowiec, Poland

**Keywords:** *Rhodiola* L., maximum entropy approach (MaxEnt), bioclimatic variables, national protected plants

## Abstract

*Rhodiola* L. has high nutritional and medicinal value. Little is known about the properties of its habitat distribution and the important eco-environmental factors shaping its suitability. *Rhodiola coccinea* (Royle) Boriss., *Rhodiola gelida* Schrenk, *Rhodiola kirilowii* (Regel) Maxim., and *Rhodiola quadrifida* (Pall.) Fisch. et Mey., which are National Grade II Protected Plants, were selected for this research. Based on high-resolution environmental data for the past, current, and future climate scenarios, we modeled the suitable habitat for four species by MaxEnt, evaluated the importance of environmental factors in shaping their distribution, and identified distribution shifts under climate change scenarios. The results indicate that the growth distribution of *R. coccinea, R. kirilowii*, and *R. quadrifida* is most affected by bio10 (mean temperature of warmest quarter), bio3 (isothermality), and bio12 (annual precipitation), whereas that of *R. gelida* is most affected by bio8 (mean temperature of wettest quarter), bio13 (precipitation of wettest month), and bio16 (precipitation of wettest quarter). Under the current climate scenario, *R. coccinea* and *R. quadrifida* are primarily distributed in Tibet, eastern Qinghai, Sichuan, northern Yunnan, and southern Gansu in China, and according to the 2070 climate scenario, the suitable habitats for both species are expected to expand. On the other hand, the suitable habitats for *R. gelida* and *R. kirilowii*, which are primarily concentrated in southwestern Xinjiang, Tibet, eastern Qinghai, Sichuan, northern Yunnan, and southern Gansu in China, are projected to decrease under the 2070 climate scenario. Given these results, the four species included in our study urgently need to be subjected to targeted observation management to ensure the renewal of *Rhodiola* communities. In particular, *R. gelida* and *R. kirilowii* should be given more attention. This study provides a useful reference with valuable insights for developing effective management and conservation strategies for these four nationally protected plant species.

## 1. Introduction

Climate change poses one of the most significant threats to global biodiversity and has a profound impact on the geographical distribution patterns of species [1,2,3]. According to the climate change report by the Intergovernmental Panel on Climate Change (IPCC), the last three decades have most likely been the warmest in the past 800 years. The report also suggests that global warming could rise to 1.5 °C above pre-industrial levels between 2030 and 2052 [4,5]. Global warming causes a rise in the temperature of the troposphere, which in turn results in a gradual increase in rainfall in the Northern Hemisphere over time. In many regions around the world, there has been a notable increase in rainfall, as well in the intensity and frequency of rainfall. The frequency of cold days and nights in both the Northern and Southern Hemispheres is decreasing, while the frequency of hot days and nights is increasing, indicating a trend towards warmer temperatures. By the end of the 20th century, annual precipitation is expected to increase in the high- and middle-latitude wet regions of the Northern Hemisphere, while decreasing in the middle and low latitudes. Climate change is primarily caused by the interaction between temperature and precipitation factors [6,7]. In particular, the rapid increase in global temperatures over the last 30 years has significantly impacted the distribution of plants due to climate change. In the future, global warming is expected to shift the spatial distribution pattern of plant growth towards higher latitudes [8,9,10]. This shows that while plants are growing in a given current environment, it does not mean that they are confined to those environmental conditions. In order to survive and reproduce, plants are bound to adapt to the environment that suits them, through the evolution of organs and even the change of life cycle [11,12]. Relevant studies have shown that climate change will alter the climate-suitable zones for plants in the future, posing a serious threat to species diversity [13].

Currently, the most frequently utilized niche models for forecasting the potential distribution of species include Bioclim (bioclimatic prediction system) [14], Domain (domain model) [15], GARP (genetic algorithm for rule-set prediction) [16], ENFA (ecological niche factor analysis) [17], and MaxEnt (maximum entropy approach) [18]. Among them, MaxEnt is the most widely used ecological niche model [19,20,21,22,23,24,25]. It has proven to be a useful tool for predicting prehistoric geological periods and future climate scenarios [26]. Moreover, the MaxEnt model is preferred for the conservation of species, determination of pedigree geography [27], and simulation of species’ potential distribution areas [28,29,30], among other research fields. The advantage of the MaxEnt model is that it can produce accurate results even with a small sample [31,32].

*Rhodiola* L. is a hemicryptophyte in the family Crassulaceae, with approximately 90 species worldwide, which are primarily distributed in Asia, North America, and the Himalayas. Of these, there exist approximately 73 species, 2 subspecies, and 7 varieties of *Rhodiola* in China, which account for approximately 80% of the world’s total *Rhodiola* resources [33]. They are primarily distributed in northwest and southwest China, and most of them grow at an altitude of 2000–5000 m in alpine rocky areas or under shrubs [33]. Molecular phylogenetic analyses suggest that *Rhodiola* originated in the Qinghai–Tibet Plateau about 21.0 Mya and rapidly diversified beginning 12.1 Mya coincident with the uplift of the Qinghai–Tibet Plateau [34]. The genus subsequently expanded into adjacent regions, with a handful of species dispersing to other parts of the globe [35]. The evolutionary history and contemporary geographic ranges suggest that *Rhodiola* species are adapted to low temperatures, making them an ideal model for investigating the response of montane herbaceous species to climate change. The genealogical patterns of three *Rhodiola* species (*R. alsia*, *R. dumulosa* and *R. kirilowii*) of the Qinghai–Tibet Plateau have been investigated, and these studies suggested that contemporary *Rhodiola* distributions have been highly influenced by the LGM temperature [36,37,38]. Meanwhile, *Rhodiola* is a highly promising medicinal plant in both the food and medicine industries owing to its anti-hypoxic and anti-aging effects.

In recent years, *Rhodiola*’s resources have been over-exploited and -utilized, leading to the destruction of its natural resources and a significant decline in reserves. This, coupled with the plant’s restrictive growth environment, has resulted in a sharp reduction in its distribution area. In this study, MaxEnt 3.4.1 software is utilized to simulate its dynamic geographic distribution over various time periods. The present study aims to discuss the potential distribution range and the major environmental factors affecting the distribution of different *Rhodiola* species during five periods: Last Interglacial (LIG), Last Glacial Maximum (LGM), Middle Holocene (MH), current, and future (under 2050 and 2070 climate scenarios). Our findings will provide a theoretical reference for future reproduction, resource utilization, and protection of this genus. Moreover, this study holds significant importance for maintaining species diversity and ecosystem stability in arid and semi-arid regions, particularly in the face of global climate change. The specific aims of our study are to (1) test which climatic factors underlie the distributions and range shifts of *Rhodiola* species in different periods (past, current, and future), and (2) test the evolution of suitable distribution areas of *Rhodiola* in different periods.

## 2. Data and Methods

### 2.1. Geographical Distribution Data for Rhodiola L.

The distribution data for four *Rhodiola* L. species were obtained from the China Virtual Herbarium (CVH, http://www.cvh.ac.cn/, accessed on 15 March 2023), National Specimen Information Infrastructure (NSII, http://www.nsii.org.cn/, accessed on 20 March 2023), and Global Biodiversity Information Facility (GBIF, https://www.gbif.org/, accessed on 26 March 2023). In this study, *Rhodiola coccinea* (Royle) Boriss., *Rhodiola gelida* Schrenk, *Rhodiola kirilowii* (Regel) Maxim., and *Rhodiola quadrifida* (Pall.) Fisch. et Mey., all of which are National Grade II Protected Plants, are selected as research objects. A total of 886 data points on the species’ natural distribution were collected. Duplicate and erroneous samples were removed. To minimize the sampling error, only one distribution point was selected within each 10 km × 10 km grid, resulting in a total of 330 distribution recording points. Of these, 77, 18, 162, and 73 belonged to *R. coccinea*, *R. gelida*, *R. kirilowii*, and *R. quadrifida*, respectively (Figure 1).

### 2.2. Environmental Parameters

In this study, we modeled the distribution patterns of the four *Rhodiola* L. varieties across past, current, and future climate scenarios. For the current climate scenarios, we initially selected 19 bioclimatic variables (Table 1) with a spatial resolution of 30 s (approximately 1 km) from the World Climate Database (http://www.worldclim.org/, accessed on 17 April 2023) [39]. For the past climate scenarios, climate change modeling data acquired from the World Climate Database (http://www.worldclim.org/, accessed on 23 April 2023) [39] was utilized based on the following criteria: MH with a 30 s spatial resolution, LGM with a 2.5 min spatial resolution, and LIG with a 30 s spatial resolution. For the future climate scenarios, we utilized climate change modeling data, downloaded from the World Climate Database, based on the representative concentration pathway scenarios: RCP8.5-2050 and RCP8.5-2070 [39].

Typically, climatic factors primarily include precipitation and temperature. There exists a certain correlation between climatic factors that may result in overfitting of forecasts. ArcGIS 10.8 software was utilized to extract climatic variable information from the distribution points of *Rhodiola* L. Furthermore, the correlation of climatic variables was analyzed using the Pearson correlation coefficient method in SPSS 23.0, based on the following criteria: If the coefficient was less than 0.8, the bioclimatic variables were retained. When the Pearson correlation coefficient between two variables exceeded 0.8, the variable with the higher contribution rate was retained. Finally, the following eight bioclimatic variables were selected: bio2, bio3, bio4, bio8, bio10, bio12, bio13, bio16.

### 2.3. Model Simulation

MaxEnt software (version 3.4.1; http://www.cs.princeton.edu/wschapire/Maxent/; accessed on 17 April 2023) was utilized for modeling. MaxEnt was independently able to perform and predict comparatively well against an ensemble approach that combined many well-used, highly regarded algorithms to identify important areas for the suitable area distribution of plants. Such findings do not necessarily imply that MaxEnt is a better technique than other approaches, and there are still cases where it is less appropriate [40]. But when modeling species distributions from incomplete data, MaxEnt should still be considered one of the most reliable and accessible technologies [41,42,43]. The first key to MaxEnt’s success is its regularization process to avoid overfitting, especially when using small sample sizes [44,45,46]. MaxEnt is able to extract useful information successfully even from incomplete data, and hence captures non-linear, complex interactions and relationships [44,45,46]. MaxEnt is not sensitive to variation in sample size [47]. Secondly, MaxEnt has been shown to be relatively insensitive to moderate sampling bias [44,46]. Although studies have shown signs of spatial bias [48], Graham et al. [31] found that MaxEnt was one of the techniques not strongly influenced by spatial errors in sampling.

The MaxEnt model approaches the problem by translating predictions of species distribution into a probabilistic model. The randomness of the species distribution prediction is made into a probability distribution and the optimal distribution probability is found [49]. The calculated results of entropy increase with the input of environmental factors related to each distribution’s data and the number of iterations. Finally, the state with the maximum entropy is obtained, that is, the state closest to the real thing. Mathematically, given a random variable *ε*, it has n different possible outcomes. X1, X2, … Xn happens with probability *p*_1_, *p*_2_…, then, the entropy *p_n_* of *ε* can be written as follows:$$H(\varepsilon )=\sum_{i=1}^{n}p_{i}\log\frac{1}{p_{i}}=-\sum_{i=1}^{n}p_{i}\log p_{i}$$

The bioclimatic variables in ASCII format, along with the latitude and longitude information for the distribution points of the four species in CSV format, were imported into MaxEnt 3.4.1 for analysis. Furthermore, 25% of the distribution points formed the test set, while the remaining 75% of the distribution points formed the training set. The prediction process was repeated 10 times. In addition, the accuracy of the prediction results was evaluated using the area under the curve (AUC) value [50], which is determined by the area enclosed by the receiver operating characteristic (ROC) curve and the horizontal axis [51]. The AUC value ranges from 0 to 1; the closer the value is to 1, the stronger is the correlation between bioclimatic variables and the predicted geographical distribution area of species, indicating the higher accuracy of the model prediction results. Different AUC values describe the accuracy of prediction results as follows: 0.50–0.60, very poor; 0.60–0.70, poor; 0.70–0.80, fair; 0.80–0.90, good; and 0.90–1.00, excellent [52].

### 2.4. Importance Assessment of Environmental Variables and Classification of Suitable Habitat

The MaxEnt model utilizes the Jackknife method to assess the contribution rate of each environmental variable to the prediction outcomes. This approach helped identify the primary ecological factors limiting the distribution of *Rhodiola* in China. The ASCII files obtained from the operation were imported into ArcGIS 10.8 and converted into raster data. The manual classification method in the Reclass tool was used to categorize the habitats based on their suitability indices, as follows: unsuitable (0–0.2), poorly suitable (0.2–0.4), moderately suitable (0.4–0.7), and highly suitable (0.7–1) habitats. Finally, the distribution and occurrence areas of the four species in China were obtained [53].

## 3. Results

### 3.1. Major Climatic Factors Affecting the Distribution of Rhodiola L.

Based on the ROC curve analysis of the AUC, the MaxEnt model for *Rhodiola* yielded AUC values ranging from 0.86 to 0.93 across different time periods. These results indicate that MaxEnt can accurately simulate the distribution range of the four *Rhodiola* species across the six time periods. Four species, with a cumulative occurrence rate of 70%, were selected 24 times in the results of the distribution models of the four species (Table 2).

Over the course of six different periods, *R. coccinea* exhibited bio3 and bio10 six times; moreover, bio4 and bio12 appeared four and two times, respectively. These four climatic variables contributed the most to species distribution, implying that compared to precipitation, temperature is more influential in determining the distribution of *R. coccinea*. Moreover, bio2, bio8, bio13, and bio16 had the least impact on *R. coccinea* distribution.

During the six different periods, *R. gelida* exhibited bio13 and bio16 six times, bio8 four times, and bio2 two times. These four climatic variables demonstrated the highest contribution to species distribution, indicating the greater influence of precipitation on *R. gelida* distribution compared to temperature. In contrast, the climatic variables bio3, bio4, bio10, and bio12 least affected the distribution of *R. gelida.*

Furthermore, bio3, bio10, and bio12 appeared six times in *R. kirilowii* over the course of six different periods. These three climatic variables contributed the most to species distribution, implying the coupling effect of temperature and precipitation on the distribution of *R. kirilowii.* In contrast, the climatic variables bio2, bio4, bio8, bio13, and bio16 demonstrated the least impact on *R. kirilowii* distribution.

Over the course of six different periods, the climatic variables bio10, bio3, and bio4 were observed six times, with a contribution rate of approximately 60%; four times; and two times, respectively, in *R. quadrifida*. These three climatic variables contributed most significantly to *R. quadrifida* distribution, suggesting that the distribution of *R. quadrifida* is primarily affected by temperature. In contrast, the climatic variables bio2, bio8, bio12, bio13, and bio16 had the least impact on *R. quadrifida* distribution.

### 3.2. Potential Geographical Distribution of Rhodiola across Different Periods

The modeling results for the distribution range of four *Rhodiola* species under six climate scenarios are presented in Figure 2 and Figure 3. The figures show all the distribution areas for *Rhodiola* in China, which include both the highly and moderately suitable areas. Under past climate scenarios (LIG, LGM, and MH), suitable habitats for *R. coccinea* were widely distributed in Tibet, Sichuan, northern Yunnan, Qinghai, and Gansu, with sporadic occurrences in western Xinjiang. Compared with MH, the current climate scenario does not demonstrate a significant change in habitats suitable for *R. coccinea* distribution. However, under the 2050 and 2070 climate scenarios, the habitat suitable for *R. coccinea* distribution is projected to shift southwards. Suitable habitats for *R. gelida* are currently widely distributed in southwest Xinjiang, northwest Tibet, Qinghai, and along the border between Gansu and Xinjiang, and compared to that under the MH, the suitable habitat for *R. gelida* distribution has changed only slightly under the current climate scenario. Under the climate scenarios for 2050 and 2070, the most suitable habitat for *R. gelida* distribution is expected to be primarily in southwest Xinjiang and northwest Tibet. These results suggest that the area of high suitability has significantly decreased. Currently, the most suitable habitats for *R. kirilowii* are widely distributed in the Qinghai–Tibet Plateau, northwest Sichuan, northeast Tibet, southeast Qinghai, northern Yunnan, and southern Gansu, and no significant change for *R. kirilowii* distribution was demonstrated compared with MH. Moreover, suitable habitats for *R. kirilowii* distribution under the climate scenarios for 2050 and 2070 exhibit no evident deviation. The suitable habitats for *R. quadrifida* are presently widely distributed in northwestern Sichuan, eastern and southern Tibet, eastern Qinghai, southern Gansu, and southern Xinjiang, and the suitable habitat range for *R. quadrifida* distribution has extended further northeast compared to that during MH. Under the 2050 and 2070 climate scenarios, the optimal distribution area for *R. quadrifida* is estimated to be primarily concentrated in Sichuan and Tibet, and that of Qinghai is expected to reduce (Figure 2), relatively.

### 3.3. Spatial Pattern Changes in Potential Suitable Areas for Rhodiola Distribution during Different Periods

The spatial patterns of the historically suitable areas and the currently potentially suitable areas for the distribution of the four selected *Rhodiola* species were compared and analyzed (Table 3). The results indicate that the suitable distribution area of *R. coccinea* decreased during the LGI-LGM period, and the loss area was estimated to be 21,369 km^2^, which accounts for 1.3% of the current suitable area for *R. coccinea* distribution. These areas of loss are mainly distributed in the northwest region of China, specifically in western Xinjiang, western Tibet, eastern Qinghai, and southern Gansu. The suitable distribution area of *R. gelida* showed an expanding trend, with the increased area being 9776 km^2^, which accounts for 1.6% of its current suitable area. The expanded suitable areas are primarily concentrated in southwest Xinjiang and western Tibet. The suitable distribution area of *R. kirilowii* decreased by 1621 km^2^, which accounts for 0.18% of the total suitable area of the species in China. The suitable distribution area of *R. quadrifida* underwent significant changes, with a reduction of 82,301 km^2^, which accounts for 4.5% of its current suitable area. These areas of loss are mainly distributed in Qinghai and Tibet.

During the LGM–MH period, the suitable distribution area of *R. gelida* and *R. kirilowii* decreased significantly, and the total areas lost were 19,129 km^2^ and 15,919 km^2^, which account for 3.2% and 1.7% of their current suitable areas, respectively. The suitable areas of loss are primarily distributed in Tibet.

During the MH–current period, the suitable distribution area of *R. coccinea* and *R. quadrifida* exhibited a decreasing trend. Specifically, the suitable distribution area loss for *R. coccinea* and *R. quadrifida* was 16,887 km^2^ and 67,614 km^2^, respectively, which account for 1.0% and 3.7% of their current suitable areas, respectively. Moreover, the suitable distribution area of *R. gelida* and *R. kirilowii* showed an expanding trend. The suitable distribution area of *R. gelida* was relatively large, covering 51,448 km^2^, which accounts for 8.5% of its current suitable area. The suitable distribution area of *R. kirilowii* increased by 13,832 km^2^, accounting for 1.5% of its current suitable area in China. These expanded suitable areas are primarily distributed in Qinghai and Tibet.

In the scenario of future climate change, variations in the spatial pattern of *Rhodiola* were compared and analyzed. The results indicate that as global warming intensifies, the majority of *Rhodiola*’s current suitable areas will remain intact under two potential climate change scenarios in the future, with retention rates exceeding 80%. Under the 2050 climate scenario, the suitable distribution areas of *R. gelida*, *R. kirilowii*, and *R. quadrifida* are expected to decrease. The suitable distribution area of *R. gelida* is estimated to decrease to 37,555 km^2^, which accounts for 6.2% of its current suitable area, with Xinjiang and Tibet being the primary distribution areas. The suitable distribution area of *R. kirilowii* is expected to reduce to 28,029 km^2^, which accounts for 3.0% of its current suitable area, with Sichuan and Tibet being the primary distribution areas. The loss in suitable distribution area of *R. quadrifida* is expected to be 11,680 km^2^, which accounts for 0.6% of its current suitable area. These areas of loss are expected to be primarily concentrated in Qinghai. Under the 2070 climate scenario, the suitable distribution areas of *R. coccinea* and *R. quadrifida* are expected to expand but those of *R. gelida* and *R. kirilowii* are estimated to show a decreasing trend, leading to a loss of 51,588 km^2^ and 40,333 km^2^, respectively, which account for 8.5% and 4.4% of their current suitable distribution areas, respectively. Newly discovered and lost suitable areas for *Rhodiola* distribution are particularly vulnerable to the impacts of climate change. Therefore, it is imperative to prioritize these areas and develop effective conservation strategies to mitigate the effects of climate change on *Rhodiola* species.

## 4. Discussion

### 4.1. Potential Distribution of Rhodiola

The response of species to rapid climate change typically involves diffusion into new adaptation zones, in situ adaptation, or extinction [54]. However, most alpine plants have limited dispersal over long distances [55,56]. *Rhodiola* primarily grows in the Qinghai–Tibet Plateau and the Hengduan Mountain area, which is consistent with the findings of the present study. Under the current climate scenario, the suitable areas for the distribution of the four studied species are mainly located in Xinjiang, Tibet, Qinghai, Sichuan, Gansu, and Yunnan. The potential distribution area of *R. gelida* in China is 1,616,100 km^2^, which accounts for 6.3% of China’s total area. The highly suitable areas, with suitability values of ≥0.7, are mainly located in southwest Xinjiang and northwest Tibet. The potential distribution areas of *R. coccinea*, *R. quadrifida*, and *R. kirilowii* in China are 605,739 km^2^, 925,245 km^2^, and 1,818,202 km^2^, which account for 16.8%, 18.9%, and 9.6% of the total land area in China, respectively. The highly suitable areas, with suitability values of ≥0.7, are mainly concentrated in Tibet, Sichuan, Qinghai, Gansu, and Yunnan.

### 4.2. Relationship between Rhodiola and Climatic Variables

Precipitation and temperature are two crucial climatic variables that impact species distribution [57,58]. Owing to China’s complex topography, there are significant variations in rainfall and temperature across different spatial scales [59]. According to the results of the analysis on the contribution rate of ecological factors, bio10 had contribution rates of 44.1% and 61.1% for *R. coccinea* and *R. quadrifida*, respectively. This suggests that the mean temperature of the warmest quarter is a decisive factor that affects the distribution of *R. coccinea* and *R. quadrifida.* Moreover, bio13 and bio16 exhibited the highest contribution rates among ecological factors, both at 31.0%. This suggests that precipitation played a significant role in determining the distribution of suitable areas for *R. gelida*. Moreover, bio10 and bio12 had the largest contribution rates to the growth of *R. kirilowii*, accounting for 31.3% and 30.9%, respectively. This indicates that temperature and precipitation jointly affect the distribution of suitable areas for *R. kirilowii* growth.

### 4.3. Spatial Distribution of Rhodiola under Climate Change

In this study, the MaxEnt model was utilized to demonstrate the potential distribution areas of four *Rhodiola* species under past, current, and future climate scenarios. The results indicate that the range of ecologically suitable areas for the four *Rhodiola* species would vary under different climatic scenarios. During LIG–MH, the suitable habitat range of all the studied species decreased, except for that of *R. quadrifida*, which increased. Under the current climate scenario, the suitable habitat range for *R. coccinea* and *R. quadrifida* has decreased, while that of *R. gelida* and *R. kirilowii* has expanded. In the future climate context, we estimate that the suitable habitat range for *R. coccinea* and *R. quadrifida* will increase under a higher concentration of greenhouse gas emissions under the RCP8.5 scenario. This implies that temperature induces a positive effect on *R. coccinea* and *R. quadrifida* growth by accelerating phenological processes and extending the growing season. Moreover, the suitable habitat range for *R. gelida* and *R. kirilowii* will gradually decrease.

## 5. Conclusions

(1) Based on the MaxEnt model, eight climate factors were selected to simulate and predict the geographical distribution and occurrence areas of four *Rhodiola* species across two temporal and spatial scales. The AUC value of the model ranged between 0.86 and 0.93, indicating the high accuracy of prediction results.

(2) Among the various climatic variables, temperature and precipitation are the primary climatic factors limiting the distribution of the four studied species.

(3) Under the current scenario, *R. coccinea* and *R. quadrifida* are primarily distributed in Tibet, eastern Qinghai, Sichuan, northern Yunnan, and southern Gansu. Under the 2070 scenario, the suitable habitat areas for both species are expected to expand. Moreover, *R. gelida* and *R. kirilowii* in China are presently mainly concentrated in southwest Xinjiang, Tibet, eastern Qinghai, Sichuan, northern Yunnan, and southern Gansu, and under the 2070 scenario, the suitable habitat areas for both species are expected to decrease. The results of this study will serve as a valuable reference for developing management and conservation strategies for the four nationally protected species of *Rhodiola.*

## Figures and Tables

**Figure 1 plants-12-03735-f001:**
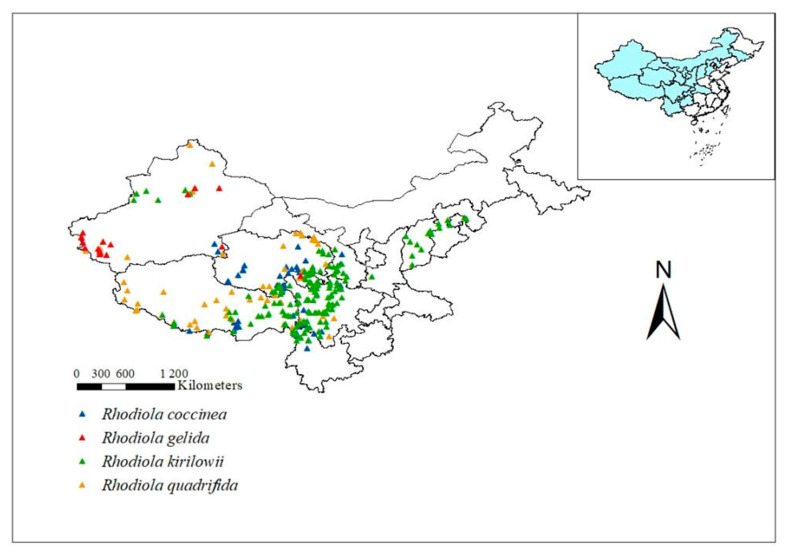
Distribution records of *R. coccinea*, *R. gelida*, *R. kirilowii* and *R. quadrifida* in China.

**Figure 2 plants-12-03735-f002:**
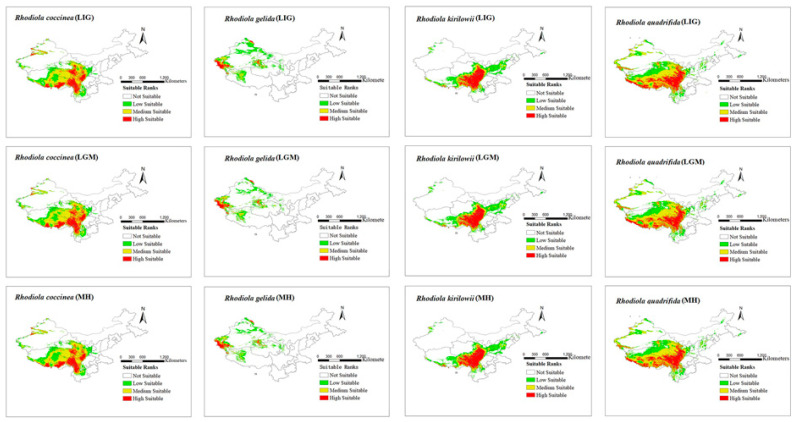
Distribution area of four species of *Rhodiola* L. under past climate scenarios.

**Figure 3 plants-12-03735-f003:**
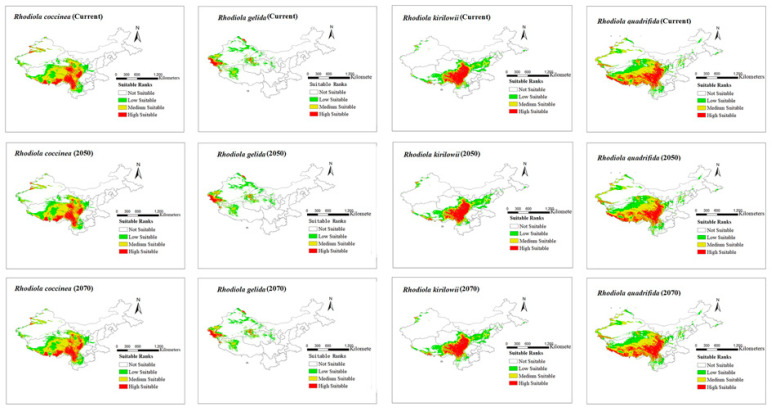
Distribution area of four species of *Rhodiola* L. under current and future climate scenarios.

**Table 1 plants-12-03735-t001:** Bioclimatic variables used in the research.

Variable	Description	Variable	Description
Bio1	Annual Mean Temperature	Bio11	Mean Temperature of Coldest Quarter
Bio2	Mean Diurnal Range (mean of monthly (max temp − min temp))	Bio12	Annual Precipitation
Bio3	Isothermality (bio2/bio7) (×100)	Bio13	Precipitation of Wettest Month
Bio4	Temperature Seasonality (standard deviation × 100)	Bio14	Precipitation of Driest Month
Bio5	Max Temperature of Warmest Month	Bio15	Precipitation Seasonality (coefficient of variation)
Bio6	Min Temperature of Coldest Month	Bio16	Precipitation of Wettest Quarter
Bio7	Temperature Annual Range (bio5–bio6)	Bio17	Precipitation of Driest Quarter
Bio8	Mean Temperature of Wettest Quarter	Bio18	Precipitation of Warmest Quarter
Bio9	Mean Temperature of Driest Quarter	Bio19	Precipitation of Coldest Quarter
Bio10	Mean Temperature of Warmest Quarter		

**Table 2 plants-12-03735-t002:** Validation of model and contribution rate of 8 different bioclimatic (bio) variables (%).

Species	AUC Data	Bio2	Bio3	Bio4	Bio8	Bio10	Bio12	Bio13	Bio16
*R. coccineas*									
LIG	0.898		15.3	16.6		44.6			
LGM	0.89		18.9	14.5		43.7			
MH	0.897		19.7	13.9		42.5			
Current	0.896		19.5	13.6		44.1			
2050	0.899		21.5			42.6	12.1		
2070	0.899		22.6			43.3	13.6		
*R. gelida*									
LIG	0.905				18.2			25.2	35.5
LGM	0.909				17.3			30.5	33
MH	0.917				18			21.1	39.4
Current	0.925				18.3			31	31
2050	0.93	17.6						27.8	34
2070	0.928	17.8						23.3	38.3
*R. kirilowii*									
LIG	0.917		24.4			31.6	31.1		
LGM	0.917		21.8			33.3	31.4		
MH	0.915		20			34.9	31.6		
Current	0.916		23			31.3	30.9		
2050	0.915		17.2			36.9	31.8		
2070	0.918		16.3			37.9	30.4		
*R. quadrifida*									
LIG	0.86		11.7			64			
LGM	0.865			13.1		61.3			
MH	0.861		12.5			61.6			
Current	0.866		13.3			61.1			
2050	0.869		14.1			57.7			
2070	0.862			15.7		62.6			

Note: only contribution rates of variables with a cumulative rate of more than 70% are shown. AUC means area under ROC.

**Table 3 plants-12-03735-t003:** Area statistics for changes in species distribution of the four *Rhodiola* L. species in six time periods (km^2^).

Species	Suitability Ranks	Species Distribution Area (km^2^)										
		LIG	LGM	Area change (LIG–LGM)	MH	Area change (LGM–MH)	Current	Area change (MH–Current)	2050	Area change (Current–2050)	2070	Area change (Current–2070)
*R.* *coccinea*	Not suitable	7,375,605	7,389,565	13,960	7,376,330	−13,235	7,403,989	27,659	7,404,598	609	7,386,625	−17,364
	Low suitable	588,549	595,958	7409	609,504	13,546	598,732	−10,772	572,518	−26,214	578,296	−20,436
	Medium suitable	1,100,264	108,094	−19,324	1,065,932	−15,008	1,066,198	266	1,144,134	77,936	1,147,899	81,701
	High suitable	554,403	552,358	−2045	567,056	14,697	549,902	−17,153	497,570	−52,332	506,001	−43,901
*R.* *gelida*	Not suitable	8,104,089	7,969,775	−134,314	8,129,178	159,403	7,956,644	−172,534	8,090,339	133,695	8,111,319	154,675
	Low suitable	951,087	1,075,625	124,538	935,351	−140,274	1,056,437	121,086	960,297	−96,140	953,351	−103,086
	Medium suitable	324,399	339,417	15,019	318,133	−21,284	371,617	53,484	327,067	−44,550	318,397	−53,221
	High suitable	239,247	234,004	−5243	236,158	2155	234,122	−2036	241,117	6995	235,755	1633
*R.* *kirilowii*	Not suitable	8,014,315	8,044,422	30,107	8,028,000	−16,422	8,018,666	−9334	8,003,861	−14,805	8,045,777	27,111
	Low suitable	675,553	647,067	−28,487	679,407	32,340	674,909	−4498	717,743	42,834	688,131	13,222
	Medium suitable	395,025	393,569	−1456	397,809	4240	402,633	4824	372,663	−29,971	364,933	−37,700
	High suitable	533,928	533,763	−165	513,604	−20,159	522,612	9008	524,554	1942	519,979	−2633
*R.* *quadrifida*	Not suitable	6,529,197	6,588,593	59,396	6,581,346	−7247	6,602,484	21,138	6,678,273	75,789	6,592,834	−9650
	Low suitable	1,128,673	1,151,578	22,905	1,151,659	81	1,198,135	46,476	1,134,026	−64,109	1,184,936	−13,199
	Medium suitable	1,092,051	1,100,747	8696	1,094,656	−6091	1,015,322	−79,334	1,042,712	27,390	1,053,106	37,784
	High Suitable	868,899	777,903	−90,997	791,160	13,257	802,880	11,720	763,810	−39,070	787,944	−14,936

## Data Availability

All generated data are included in this article.

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
