# Peer review of "Predicting the Potential Geographical Distribution of Rhodiola L. in China under Climate Change Scenarios"

_plants, 2023, doi:10.3390/plants12213735_

Round 1

Reviewer 1 Report

Comments and Suggestions for Authors

The manuscript entitled: Predicting the potential geographical distribution of Rhodiola L. in China under climate change scenarios. It is an interesting paper on the use of relatively new tools for ecological niche modelling such as MaxEnt. The results could be interesting, but the manuscript leaves many questions unanswered before it can be published in a journal such as Plants.

Introduction.

The introduction should provide sufficient bibliographical background to robustly support the development of your manuscript. In this sense, the introduction is too general, it talks a little about the different IPCC scenarios and something about the distribution of Rhodiola. But it does not go into the more ecological aspects of the genus, let alone the ecology, bioclimatology and biogeography of each of the species studied. On the other hand, although in this type of work the objectives are obvious "To use a software/algorithm and a database to model the distribution of a taxon..." in this case, using the software to generate future climate scenarios, the objectives are not entirely clear. What interest is there, or what new science do the authors bring to the table? Such approaches, with the current WorldClim + GBI databases, could lead to modelling for each of the species described. With this same methodology, more than 300,000 publishable papers like the present manuscript could be done from the home office without leaving home or contributing new distribution data, or other field work, which would add real value to this type of approach. 

On the other hand, this type of study, if not carried out and designed well, can rarely be representative of the ecological reality of a species.

The starting hypothesis is neither implicitly nor explicitly stated, this reviewer intuits that the starting hypothesis is that there are climatic variables that determine the distribution of the different Rhodiola species under study, and that the change in these climatic variables will affect the distribution projection in future scenarios...

M&M

There are many methodological gaps that need to be filled. It is not clear how the maximum entropy algorithm (MaxEnt) works in the case of your species. This type of algorithm has a big drawback, which is that it is very easy to use and produces very nice graphs, but it is very difficult to understand how it works and how our data leads to the predictions. The AUC is not very representative and is only useful for comparing models of the same species with each other, comparing AUCs of different species is not entirely correct. 

Yackulic, C. B., Chandler, R., Zipkin, E. F., Royle, J. A., Nichols, J. D., Campbell Grant, E. H., & Veran, S. (2013). Presence-only modelling using MAXENT: when can we trust the inferences?. Methods in Ecology and Evolution, 4(3), 236-243.

Other aspects are present in the attached file as commentary to the text.

Results.

The results, a priori seem coherent, but without a robust methodology, they can hardly be realistic and reliable to draw a solid conclusion.

See comments in the attached file.

Reviewer 2 Report

Comments and Suggestions for Authors

This study has been very carefully conducted, using up to date methods.  The presentation is also very clear.  

I think it would be worth adding a cautionary note, relevant to all such studies:  The physical environment where plants currently grow obviously indicates that they can tolerate that environment.  However, it does not mean that they are restricted to, or even would do best in those environmental conditions.  There can be multiple other reasons for the absence of a species in a given environment, which may be unrelated to tolerance of the physical environment. 

Round 2

Reviewer 1 Report

Comments and Suggestions for Authors

Minor comments:

Please try not to repeat keywords in the title of the manuscript.

Lines 111-113: Please check the correct formatting of the text of the Rhodiola species.

Lines 179-183: Are the different AUC values (i.e. the ranges: 0.50-0.60, very poor; 0.60-0.70, poor; 0.70-0.80, fair; 0.80-0.90, good; and 0.90-1.00, excellent) proposed by the authors, or have the authors used any literature? Please clarify this issue.

Figure 2 has a very low quality, it does not show well. In addition, the photocomposition is very poor and sloppy, the margins and lines between images are misaligned. It would be better to divide this image into two parts, or put it in an appendix or supplementary material. 

Table 3 is not well laid out and integrated into the body of the text, and it is not complete. Please remap the table and resize it to the correct size.
